

# Seasonal climate influences on the timing of the Australian monsoon onset

Joel Lisonbee[1], Joachim Ribbe[1]

[1]School of Sciences, University of Southern Queensland, Toowoomba, 4350, Australia

*Correspondence to*: Joel Lisonbee (joel.lisonbee@usq.edu.au)

**Abstract.** The timing of the first monsoon burst of the season, or the monsoon onset, can be a critical piece of information for agriculture, fire management, water management and emergency response in monsoon regions. Why do some monsoon seasons start earlier or later than others? Previous research has investigated the impact of climate influences such as the El Niño–Southern Oscillation (ENSO) on monsoon variability, but most studies have considered only the impact on rainfall and
not the timing of the onset. While these questions could be applied to any monsoon system, this research presented in this paper has focused on the Australian monsoon. Even with the wealth of research available on the variability of the Australian monsoon season, the timing of the monsoon onset is one aspect of seasonal variability that still lacks skilful seasonal prediction. To help us better understand the influence of large-scales climate drivers on monsoon onset timing, we recreated 11 previously published Australian monsoon onset datasets and extended these to all cover the same period from the 1950-51 through the
2020-21 Australian wet seasons. The extended datasets were then tested for correlations with several standard climate indices to identify which climate drivers could be used as predictors for monsoon onset timing. The results show that many of the relationships between monsoon onset dates and ENSO that were previously published are not as strong when considering the extended datasets. Only a strong La Niña pattern usually has an impact on monsoon onset timing, while ENSO–neutral and El Niño patterns lacked a similar relationship. Detrended Indian Ocean Dipole (IOD) data showed a weak relationship with
monsoon onset dates, but when the trend in the IOD data is retained, the relationship with onset dates diminishes. Other patterns of climate variability showed little relationship with Australian monsoon onset dates. Since ENSO is a tropical climate process with global impacts, it is prudent to further re-examine its influences in other monsoon regions too, with the aim to evaluate and improve previously established prediction methodologies.

## 1 Introduction

The livelihood of about 50% to 60% of the world's population is impacted by the global monsoon system (e.g., Qiao et al., 2012; Rajan et al., 2005; Wang and Ding, 2008; Yancheva et al., 2007). The monsoon is generally understood to be the seasonal change from dry to wet along with a reversal of the prevailing winds (Ramage, 1971). Monsoonal climates are characterised by dry winters followed by very wet summers when, by at least one definition, over 70% of the annual rainfall accumulates (CLIVAR, 2015; Qian et al., 2002; Zhang and Wang, 2008). With all regional monsoon systems, the seasonal monsoon onset,
or the first burst of monsoon rains of the season, is a much-anticipated event with documented temporal variability (Ali et al., 2020; Fitzpatrick et al., 2015; Lisonbee et al., 2020; Parija, 2018; Pradhan et al., 2017). This variability may be driven by





larger-scale climate variability, such as the El Niño Southern Oscillation (ENSO). Correlations between ENSO and the monsoon onset have been reported for the South Asian and East Asian monsoons (Wang et al., 2008b; Zhou and Chan, 2007), the Indian monsoon (Misra et al., 2018; Misra and Bhardwaj, 2019; Noska and Misra, 2016), the African and south African
monsoons (Semazzi et al., 2015), the south American monsoon (Grimm et al., 2015), the Mexican and southwest U.S. monsoon (Gochis, 2015), and the Australian monsoon (Drosdowsky, 1996; Holland, 1986; Kajikawa et al., 2010; Lisonbee et al., 2020). However, for the Australian monsoon the literature does not completely agree on the degree of influence had by ENSO.

The Drosdowsky (1996) monsoon onset definition  is often used as a standard for Australian monsoon onset research (e.g., Berry and Reeder, 2016; Davidson et al., 2007; Evans et al., 2014; Kajikawa et al., 2010; Kim et al., 2006; Pope et al., 2009;
Wheeler and Hendon, 2004; Zhang and Wang, 2008) and its practical application in an operational environment in the Darwin Regional Forecast Centre (Shaik and Lisonbee, 2012). Drosdowsky (1996) defined the Australian monsoon as a burst of westerly winds as represented by a deep layer tropospheric mean, with easterly winds aloft as measured at Darwin, Northern Territory, Australia. Using data from the 35 years from 1957/58 to 1991/92, Drosdowsky (1996) calculated a -56% correlation with the September-November (SON) Southern Oscillation Index (SOI), a measure of the state of ENSO. During those years,
there were eight El Niño and six La Niña events. Subsequently, there have been seven more El Niño events and seven more La Niña events until the 2020/21 season (see Section 5.2). The original motivation for this research was to test if the correlations reported by Drosdowsky (1996) are still valid when including 28 more years of data (i.e., seasons 1992/93-2020/21) in the calculation with the overall goal to better understand and utilise the potential predictability of the climate system based on seasonal-scale climate variations. The subsequent research questions that arose are these: firstly, can a similar correlation be
seen from other ENSO or non-ENSO indices? Drosdowsky (1996) was published before the discovery of the Indian Ocean Dipole (IOD) by Saji et al. (1999) and other climate indices, such as central Pacific sea surface temperatures (SST), rose to prominence What other seasonal-scale climate variations may be influencing the timing of the Australian monsoon onset? Secondly, this research question was extended to other Australian monsoon onset methodology. Others have defined the monsoon onset in ways that pin the "onset" to different events in the wet season (Lisonbee et al., 2020) and report varying
relationships with ENSO. For example, Kajikawa et al. (2010) reported a correlation coefficient between the SON SOI and Australian monsoon onset of -0.48 while Holland (1986) showed no significant correlation between seasonal monsoon onset and the SOI prior to the summer monsoon season. Do these correlations remain robust when more decades of data are considered? How can the relationships between monsoon onset and climate influences from different onset criteria accurately be compared when each respective dataset covers different time periods?  Therefore, the aim of this study is to further
investigate how much seasonal-scale climate drivers influence the timing of the Australian monsoon onset based on various onset criteria over the same time period.

For this analysis we will focus on monsoon onset definitions that include a dynamical component, such as a wind reversal. Lisonbee et al. (2020) categorised Australian monsoon onset definitions by those that are based on a wind-criteria and those





that are based on a rainfall-criteria. We consider "monsoon" definitions based solely on rainfall to indicate the onset date of
the wet season rains and not the dynamical monsoon. While defining and predicting the beginning of the rainy season has
useful application (Balston and English, 2009; Cook and Heerdegen, 2001; Cowan et al., 2020; Drosdowsky and Wheeler,
2014; Lo et al., 2007; Nicholls, 1984; Nicholls et al., 1982; Smith et al., 2008), understanding and predicting the onset of the
dynamical monsoon on a seasonal timescale aids our understanding of what drives variability in the monsoon weather pattern
and is thus likely to improve monsoon onset forecasting skill when using statistical models. While this study focuses on the
Australian monsoon, from a global perspective an understanding of the dynamical monsoon onset is important because:
tropical cyclones are more likely to form along the monsoon trough (Choi and Kim, 2020; Davidson et al., 1989; Mcbride,
1983; Wheeler and McBride, 2011); monsoon bursts, whether they include a tropical cyclone or not, can have serious impacts
on public health and safety (Martinez et al., 2020), transport and aviation (Pramono et al., 2020), flooding and ecological
effects (Crook et al., 2020), and the local economy (Jain et al., 2015). Even with the wealth of research available on the
variability of the Australian monsoon season, the timing of the dynamical monsoon onset is one aspect of the monsoon that
still lacks skilful seasonal prediction.

This paper presents a statistical analysis of seasonal-scale (1–6 month) climate influences on the timing of the dynamical onset
of the Australian monsoon for the period 1950-51 to 2020-21. To do this we, firstly, isolated monsoon onset definitions that:
(1) focus on northern Australia; (2) required some dynamical component (e.g. reversal of lower tropospheric winds) to
determine when the monsoon was active over the region; and (3) included a list of onset dates within the respective publications
(see Lisonbee et al. 2020). Secondly, we recreated the annual monsoon onset dates using the methods described in each of
these papers. Thirdly, we extended these monsoon onset datasets to cover the same time period (1950-51 through 2020-21).
Finally, with a standard data set for each monsoon onset definition, we computed statistical correlations with known seasonal-
scale climate drivers to investigate if the timing of the Australian monsoon onset can be reliably predicted on a seasonal
timescale [including testing the correlations reported in Drosdowsky (1996)] and also, which climate drivers and which
monsoon onset definitions provide the best predictability.

The climate indices considered in this study include: six ENSO indices, three Indian Ocean SST indices including the Dipole
Mode Index (DMI) as a measure of the IOD (Saji et al., 1999; Taschetto et al., 2011; Verdon and Franks, 2005), three polar
annular mode indices (Dai and Tan, 2017; Marshall and (Eds)., 2018; Mo, 2000; Thompson and Wallace, 1998), the
stratospheric Quasi-Biennial Oscillation (QBO), the North Atlantic Oscillation (Barnston and Livezey, 1987), the three
components of the Amundsen Sea index (Raphael et al., 2016), and three northern hemisphere monsoon indices (Wang et al.,
2001; Wang and Fan, 1999; Webster and Yang, 1992). The physical connection between the Indian monsoon and the Australian
monsoon has been investigated previously (Chang and Li, 2000; Kim and Kim, 2016; Li et al., 2001; Meehl, 1994; Meehl and
Arblaster, 2002; Pillai and Mohankumar, 2007; Stuecker et al., 2015; Suppiah, 1992; Wang et al., 2003, 2008a; Wu and Chan,
2005; Yu et al., 2003), but these studies considered seasonal rainfall and not the timing of the Australian monsoon onset, hence



we have chosen to include these in our analysis. The Madden-Julian Oscillation (MJO) was not included in this study for two reasons. First, the link between the MJO and the onset of the Australian monsoon has a been heavily investigated in the literature (most notably by Wheeler and Hendon 2004, but also by Hendon and Liebmann 1990a,b; Hendon et al. 1989a; Joseph et al. 1991; Pope et al. 2009; Wheeler and McBride 2012). Secondly, the MJO provides skilful predictability on a time-
scale of weeks (Lim et al., 2018) while the analysis presented in this paper focuses on the predictive correlations at the seasonal timescale.

## 2 Data

The data used in this research came from various sources, including the European Centre for Medium-Range Weather Forecasts (ECMWF), the National Centre for Atmospheric Research (NCAR) and the Australian Bureau of Meteorology. As much as
possible, the reproduced monsoon onset dates used the same data used in each respective publication. Where that data is no longer available, substitutions were made. The original data used and the data used in the re-creation are listed in section 2.1. The climate indices used, including their source data, are listed in section 2.2.

### 2.1 Onset data

In this study, it is important to compare all previously published monsoon onset detection methods to the same temporal period
to ensure all cover the same "events" of climate variability (i.e., the same ENSO, IOD, etc. patterns). Table 1 includes the reference of each onset criteria reproduced, the description of the meteorological data from the original source and the data used to reproduce that methodology.

Darwin sounding data was used for several monsoon onset re-creations (Drosdowsky, 1996; Holland, 1986; Troup, 1961). Sounding data was obtained from the Australian Bureau of Meteorology directly (available upon request from
http://www.bom.gov.au/climate/data-services/). One of the limitations of this study is the use of pre-1957 sounding data at Darwin. It should be noted that prior to 1957/58 the soundings timings, methods and reported heights were non-standard and sometimes irregular (Ramella Pralungo et al., 2013). The inclusion of this data means that onset dates that were calculated using sounding data pre-1957 should be used and interpreted with caution.

### 2.2 Climate data

The data sources for seasonal climate indices that were used to check for correlations in this study are listed in Table 2.

## 3 Method

The method to test the correlations of monsoon onset criteria with seasonal climate influences took four steps:

The first step was to isolate Australian monsoon onset definitions that fit the scope and desired outcomes of this project. We required some dynamical component (e.g. reversal of lower tropospheric winds) to determine when the monsoon was active over the region. A final limit in the scope of this work was to re-create only those works that included a list of onset dates within the respective publications so that we could test if we were re-creating the onset dates correctly (see Lisonbee et al., 2020).

The second step was to recreate the onset dates by following the methodology described in each paper. Only those works that could be sufficiently reproduced were tested for correlations with climate indices.

The third step was to extend each of the selected monsoon onset datasets to cover the same time period. The extended datasets cover the period from the 1950/51 Australian wet season to the 2020/21 season. This period is limited based on data availability where, in this study, the ECMWF ERA-5 reanalysis, used for the Hung and Yanai (2004) re-creation, begins in 1950 and, thus, presented the limit in dataset reproductions. This allowed comparisons of monsoon onset datasets that cover the same "events" of climate variability (i.e., the same ENSO, IOD, etc. patterns). For example, Kajikawa et al., (2010) show a correlation coefficient between the September-November SOI and monsoon onset of -0.48, calculated using the years 1948-2005 and included 10 La Niña events (SON SOI >0.8) while Drosdowsky (1996) showed a correlation coefficient between onset date and the SON SOI of -0.56 for 1957 through 1992 which included only six La Niña events (SON SOI >0.8). In the full period from 1950-2020, there were 11 positive IOD events (SON DMI > 0.4) and 16 negative IOD events (SON DMI <-0.4), 16 positive and 21 negative ENSO events (this count using the SON NINO3.4 SST anomalies >0.7 °C and <-0.7 °C, respectively).

The final step was to test the correlations of the monsoon onset dates with the climate indices. Each monsoon onset definition (Table 1) was paired with each of the climate indices (Table 2). Each pair was tested for normalcy. Where both pairs fit a Gaussian distribution then the Pearson coefficient ($\rho$) and corresponding two-tailed significance test value (p) were calculated. Where either of the two pairs did not fit a Gaussian distribution then the Kendall's coefficient ($\tau$) and corresponding p-value were calculated. (As a matter of convention) we have considered $p<0.05$ to represent statistical significance. Where the correlations were larger than ±0.3, we considered the monsoon onset to be somewhat influenced by the climate driver and where the correlation coefficients were larger than ±0.6 we considered the monsoon onset to be largely influenced by the climate driver and sufficient to be used as a predictive tool.

### 3.1 Troup (1961) onset definition

Troup (1961) described the Australian monsoon onset using both wind and rainfall. For a rainfall onset, Troup (1961) analysed rainfall data at six locations near Darwin for the wet seasons from 1955/56 – 1958/59. For each season in the study period, Troup defined the onset to have occurred when 4 out of the 6 stations experienced their first rainfall event simultaneously after



1 November and the area-averaged rainfall over N days exceeded 0.75(N + 1) inches (19 mm/day). Using upper-air data from Darwin airport, Troup isolated the zonal wind component at "3000 feet" (915 m) and identified spells of moderate west winds.

A westerly wind spell of *N* days occurred when the cumulative zonal component exceeds 10(N+1) knots, and ended when this component was less than five knots for two consecutive days. When extending the dataset, we considered the rainfall onset and wind onset separately, but we also noted the first day when both the wind and the rainfall criteria were met at the same time (similar to Hendon and Liebmann 1990b).

### 3.2 Murakami and Sumi (1982) onset definition

Murakami and Sumi (1982) used the enhanced observation data networks of the Global Weather Experiment and of its component experiment, the Winter Monsoon Experiment (WMONEX) to analyse the Australian monsoon. They defined monsoon onset using the mean 850 hPa zonal wind averaged along 10 °S from 100 to 180 °E; onset occurred at the first appearance of mean westerlies along this line. Murakami and Sumi (1982) provided the onset date for only one monsoon season, 1978-79. Following the Murakami and Sumi (1982) methodology, we were able to reproduce the 1978-79 onset and

extend the onset dataset using the NCEP/NCAR 40-year reanalysis data (Kalnay et al., 1996).

### 3.3 Holland (1986) onset definition

Holland (1986) defined monsoon onset as the first westerly winds at the 850 hPa level at Darwin Airport. Holland (1986) analysed seasons 1952/53 – 1982/83, and took a special focus in the 1978–79 wet season as the year of the WMONEX study. Holland averaged the daily 850 hPa level winds to remove diurnal variations and produce a daily time series. He then smoothed

out other minor variations in the data using a cubic spline method to the yearly time sequence of the daily mean winds. He was then able to analyse the onset and retreat and the burst and break periods within any season. Drosdowsky (1996) attempted to recreate the onset dates from Holland (1986), but was unable to recreate the results. To explain the differing results, Drosdowsky (1996) pointed to, and criticised, the use of a smoothed time series at a single pressure level. Drosdowsky (1996) points out examples where the smoothed single-level winds miss the actual onset events because either the winds at 850 hPa

are not representative of the lower mid-tropospheric zonal wind, or the low-pass filtering over the data blurs an abrupt change in the deep-layer winds over several days. Hendon and Liebmann (1990a) built upon the wind definition of Holland 1986 but they replaced the cubic spline with a 1-2-3-2-1 running mean to smooth out synoptic fluctuations.

### 3.4 Hendon and Liebmann (1990a) onset definition

Hendon and Liebmann published two papers on the NAM in 1990. The first (Hendon and Liebmann, 1990a) was specifically

regarding the Australian monsoon onset while the second (Hendon and Liebmann, 1990b) examined the mechanisms for the variability within the season. In these papers, Hendon and Liebmann define the onset using both wind and rainfall. The wind data is taken from the Darwin Airport upper air record and the rainfall is taken as the daily area averaged rainfall for stations





north of 15 °S in Australia. The season from 1957-58 through 1986-87 were considered. Onset was determined by the first detection of "wet westerlies" at 850 hPa—meaning area averaged rainfall of at least 7.5 mm/d coincident with the wind criteria

adopted from Holland (1986) but filtered with a 1-2-3-2-1 running mean as oppose to the cubic spline filter that Holland used.

Drosdowsky (1996) attempted to re-create the Hendon and Leibmann (1990a) results without much success. He points to the lack of clarity in their description of their techniques and datasets. Drosdowsky (1996) is also very critical of the use of a filter to smooth the daily wind data, pointing to examples when the wind reversal was quite abrupt but the smoothed data produces a gradual reversal over several days.

Hendon and Liebmann (1990a) are vague on their description of the data used. For the 850 hPa zonal wind the use the upper air record at Darwin "as per Holland (1986)". In attempting to recreate these results we used all available sounding data each day to produce daily averages of the 850 hPa level winds [as per the reproduction of Holland (1986)]. We then smoothed the daily data with a 1-2-3-2-1 weighted running mean. Drosdowsky suspects that Hendon and Leibmann may have also removed the annual cycle from their daily wind dataset, although this was not mentioned in Hendon and Liebmann (1990a) methodology

(Drosdowsky, 1996). We did not attempt to remove an annual cycle.

The rainfall data used in Hendon and Liebmann (1990a) is described as "the daily record of area averaged rainfall for stations north of 15° S in Australia". They provide a list of rainfall stations used in their Table 1, but in that table Darwin Airport is the only location listed that is north of 15° S in Australia. In attempting to reproduce the Hendon and Leibmann (1990a) onset dates we used both the daily rainfall record at Darwin Airport and the gridded rainfall data for all points in Australia north of

15° S. This data was obtained from the SILO dataset ([https://www.longpaddock.qld.gov.au/silo/gridded-data/](https://www.longpaddock.qld.gov.au/silo/gridded-data/)).

### 3.5 Drosdowsky (1996) onset definition

The Drosdowsky (1996) deep layer mean westerly wind definition defines the monsoon onset at Darwin using Darwin sounding data. Drosdowsky (1996) developed definitions of active and break cycles and the onset and retreat of the monsoon. Using the 2300 UTC upper air data from Darwin Airport, the monsoon was defined as deep low-level westerly flow overlain

by strong upper-level easterlies. The mass-weighted deep layer mean winds in the lower troposphere is calculated using Eq. (1) and the mass-weighted deep layer mean winds in the upper troposphere is calculated using Eq. (2):

$$DLM_{lower} = 0.1U_{sfc} + 0.15U_{900\,hPa} + 0.12U_{850\,hPa} + 0.15U_{780\,hPa} + 0.13U_{700\,hPa} \qquad (1)$$
$$+ 0.1U_{650\,hPa} + 0.15U_{600\,hPa} + 0.1U_{500\,hPa} \;,$$





$$DLM_{upper} = 0.25U_{200\,hPa} + 0.5U_{150\,hPa} + 0.25U_{100\,hPa}\,, \tag{2}$$

where U is the westerly wind component and the subscripts indicate the pressure level of that wind measurement. Monsoon onset was considered when the average $DLM_{lower}$ over N days exceeded 2.5(N+1)/N m/s and $DLM_{upper}$ is easterly (U<0). These lower level westerly winds had to be in place for at least two consecutive days to be considered an active monsoon period and the minimum break period between bursts is three days such that westerly wind bursts separated by only one or two days were concatenated. The monsoon onset was defined as the first day of the first active monsoon period within the season.

Drosdowsky (1996) found that some subjective assessment of onset dates cannot be avoided, but the onset dates from the five years from 1987/88 to 1991/92 were determined completely objectively. The years when a subjective analysis was needed, the choices made by Drosdowsky (1996) seemed logical and where also used in the recreation. In most years when objective analysis was applied the onsets criteria was met for $DLM_{lower}$ but the upper level easterlies had not established. For some other years missing data made objective analysis impossible—one obvious example was the monsoon onset on 25 December 1974 where there is missing data from 25 to 30 December due to the passage of tropical cyclone *Tracey*. The treatment of missing data is described in Drosdowsky (1996). For the extended dataset (1992/93-2019/20) subjective analysis was applied to the 1998/99 and 2004/05 seasons.

### 3.6 Hung and Yanai (2004) onset definition

Hung and Yanai (2004) defined the onset of the Australian monsoon using the reanalyses from the European Centre for Medium-Range Weather Forecasts (ECMWF) ERA-15, outgoing long-wave radiation (OLR) and precipitation data for 1979–1993. Onset is defined as the first day with average 850 hPa zonal wind exceeding 2 m/s over a north Australian/Arafura sea domain (2–15° S, 115–150 °E) when the westerly wind is sustained for longer than 10 days and the OLR is lower than 210 W/m$^2$ for "at least several days" during the 10-day period. Using this definition, the mean onset date from the 14 years studied is 25 December with a standard deviation of two weeks.

In reproducing the Hung and Hung and Yanai (2004) onset dates the ERA-15 data was no longer available. ERA-5 and ERA-interim reanalysis data were used as a substitution, both producing the same results; ERA-5 data is shown here. The re-created onset dataset matched the original dates to within one day in all but two cases (see Figure 1) which will be discussed in the results section.



### 3.7 Davidson et al. (2007) onset definition

Davidson et al. (2007) defined the NAM onset using a wind-only criteria in a fashion similar to Drosdowsky (1996), but for a general monsoon region as opposed to the point-observations at Darwin Airport that were used in Drosdowsky (1996). Davidson et al. (2007) began with a comparison between both the NCEP and ERA-40 datasets. The close agreement in the

zonal wind and MSLP indicated consistency in the datasets for these standard variables. They concluded that either reanalyses are suitable for their purposes since the temporal changes are consistent within each dataset, and they used the NCAP dataset in their analysis of monsoon onset dates. Davidson et al. (2007) defined the monsoon onset as a sudden strengthening and deepening in tropical westerly winds, which are overlain with upper-tropospheric easterlies over a monsoonal region (15°–5°S, 110°–140°E). The Lower-tropospheric westerly winds had to meet a minimum threshold of 2.5 m/s and extend to at least

600 hPa. Easterlies in the upper troposphere must overlay the westerlies. This structure must persist for at least 4 days. The authors did not specify which pressure levels they considered to be "the lower troposphere" or which levels they consider to be "the upper troposphere". In both the reproduced and extended datasets, we used 1000-500 hPa to represent the lower troposphere and levels 250-150 to represent the upper troposphere. For all other aspects of this reproduction we were able to follow the methodology described in Davidson et al. (2007).

### 3.8 Kajikawa et al. (2010) onset definition

Kajikawa et al. (2010) derived an Australian monsoon index (AUSMI) to examine intra-seasonal variability, including the onset. This index is defined using 850 hPa zonal wind averaged over the area 5 °S–15 °S, 110 °E–130 °E using daily NCEP reanalysis data where positive values indicate a westerly wind.

Kajikawa et al. (2010) patterned their onset criteria after the Wang et al. (2004) monsoon onset definition for the South China

Sea monsoon onset. The Australian monsoon onset is defined as the first day after 1 November that satisfies the following three criteria: (1) on the onset day and during the 5 days after the onset day the averaged AUSMI must be positive; (2) the pentad mean AUSMI is positive in the at least three of the subsequent four pentads; (3) the accumulative four-pentad mean AUSMI >1m/s (Kajikawa et al., 2010).

### 3.9 Zhang (2010) onset definition

Zhang (2010) defined the Australian monsoon onset using a normalised precipitable water index similar to Zeng and Lu (2004), who created a global monsoon index based on a normalised precipitable water index. Zhang (2010) used this same index to define onset and retreat dates for northern Australia and Darwin. The normalised precipitable water (PW) index is defined as

$$PW_n = \frac{(PW - PW_{min})}{(PW_{max} - PW_{min})}$$



where PW$_{max}$ and PW$_{min}$ are the 44-year mean of daily PW maximum and minimum in each of the 44 years during the period
of 1958–2001 and at each grid point. Once PW$_n$ was calculated for each day and each grid point, they then define monsoon
onset/retreat as follows: first, they assess if the PW$_n$ exceeds 0.65 for three continuous days for at least 7 of the 9 points around
a location; then they assess whether 850 hPa monsoon westerly is established, with averaged zonal wind of the nine points
around the location remaining westerly for the same three days.

Zhang (2010) used daily and monthly ERA-40 reanalysis data for the period of 1958–2001. To test the ability to recreate the
Zhang (2010) methodology we used daily U-wind component and total column water data from ERA-40. The primary
limitation to using ERA-40 data is that it only covers the years 1958 to 2002. To extend the dataset we would need to use a
different reanalysis dataset such as ERA-interim or ERA-5. As will be shown below, we could not satisfactorily recreate the
Zhang (2010) methodology for ERA-40 and, therefore, we did not repeat the process with a longer dataset.

## 4 Results

In the following subsections are the results of each stage of this analysis. Firstly, a report on the accuracy of the re-created
monsoon onset datasets in section 4.1. Following by a detailed analysis of correlations with climate indices with the
Drosdowsky (1996) extended dataset in section 4.2. And then a summary of the same analysis for the other eight extended
monsoon onset datasets in section 4.3.

### 4.1 Monsoon onset reproductions

To answer the research questioned posed in the Introduction, we created monsoon onset data that covered a standard time
period such that correlations of the individual onset methodologies overlapped with the same climate indices. Out results in
reproducing the onset methods described in Section 3 are described here in chronological order of the respective publication
date and are also shown in Figure 1 (a-k). Re-created data are shown in comparison to the original data. Also shown are the
onset data using each definition for the extended period.

**Troup (1961)** considered the rainfall onset and the wind onset to be two separate events that occasionally overlapped. When
extending the dataset, we also considered the rainfall onset and wind onset separately, but we also noted the first day when
both the wind and the rainfall criteria were met at the same time (similar to Hendon and Liebmann 1990b). This method
successfully reproduced the onset from the four years studied by Troup (Figure 1a), but we found that, when extending the
dataset to the present, there were a few years when both criteria were not met at the same time at any point within the season.
This also provided for a few very late onset dates (e.g. February and March). While Troupe (1961) included the dates for all
monsoon "bursts"—a term Troup (1961) used to describe an active monsoon period—within each season, here we are
considering only the dates of the first burst each season as the "onset". The extended dataset captured the onset for each year
precisely, however, it did not capture the exact dates of each burst as described by Troup (1961), although the dates were off



by only a day or two. We suspect Troup (1961) used some subjective analysis in determining these dates. The extended Troup
(1961) dataset showed mean onset dates of 31 December using the rainfall criteria, 29 December using the wind criteria, and
20 January using both criteria combined, each with a standard deviation of 25 days.

**Murakami and Sumi (1982)** provided the onset date for only one monsoon season, 1978-79 (Figure 1b). Following the
Murakami and Sumi (1982) methodology, we were able to reproduce the 1978-79 onset and extend the onset dataset using the
NCEP/NCAR 40-year reanalysis data (Kalnay et al., 1996). The reconstructed dataset shows a mean onset date of 26 December
with a standard deviation of 16 days.

**The Holland (1986)** onset dates, and associated uncertainty estimates were taken from Table 3 in Lisonbee et al. (2020). We
could not recreate the Holland (1986) onset dates using a cubic spline smoothing method, experiencing similar problems as
Drosdowsky (1996). Through some experimentation we were able to recreate most of the Holland (1986) dates to within the
uncertainty estimates using 19 iterations of a 1-2-3-2-1 filter similar to Hendon and Liebmann (1990a). We recreated 22 onset
dates (79%) to within the uncertainty estimates, 5 onset dates (18%) that are less than one week outside the uncertainty
estimates, and one onset date that was more than one week outside the uncertainty estimate (Figure 1c). Holland (1986) showed
the average onset date for the 30 years from 1952/53 to 1982/83 was 24 December, with the earliest onset date of 23 November
and the latest date of 27 January. When considering the full extended dataset, the mean onset date is 22 December with a
standard deviation of 16 days. We could not recreate the earliest onset date from the original dataset, but later in the
reconstructed dataset the earliest onset date is 15 November.

For the **Hendon and Liebmann (1990a)** onset dates, only 4 of the 30 seasons (13%) were successfully reproduced, 17 seasons
(47%) were within seven days, and 9 seasons (30%) were more than 7 days away from the original Hendon and Liebmann
(1990a) dates (Figure 1d). Similar to the Drosdowsky (1996) attempt to reproduce the Hendon and Leibmann (1990a) onset
dates, we found that there were aspects of their methodology and data used that were unclear. It is possible that Hendon and
Leibmann (1990a) did not use daily averaged 850 hPa winds, perhaps they used only the 12 or 23 UTC soundings. Drosdowsky
(1996) suggests Hendon and Liebmann (1990a) may have removed the mean seasonal wind cycle from their wind data, without
mentioning this in their methodology.  It is also possible that the averaged gridded daily rainfall data we are using does not
match the areal averaged station data for stations north of 15° S in Australia. Overall, we did not consider this a successful
reproduction and the extended Hendon and Liebmann (1990a) dataset was not included in correlations calculations.

Our recreation and extension of the **Drosdowsky (1996)** onset dates is shown in Figure 1e. Our analysis reproduced the precise
onset dates for 13 of 35 years (37%), was different by one day for 17 of the 35 years (48%), and was more than one day but
less than 5 days different for 5 of the 35 years (14%). Drosdowsky (1996) included some subjective analysis in determining
the onset data, but the five years from the 1987/88 season to the 1991/92 season were  found completely by objective analysis,
we were also able to reproduce precisely for three seasons and one day different for 1988/89 and 1991/92 seasons. Drosdowsky




(1996) showed the average onset date for the 35 years from 1957/58 to 1991/92 was "28-29 December". When considering the full extended dataset, the mean onset date is 29 December with a standard deviation of 16 days.

The re-created **Hung and Yanai (2004)** onset dataset matched the original dates to within one day in all but two cases (Figure 1f). The 1983-84 and 1989-90 seasons present a very large discrepancy which is probably due to using the ERA-5 data rather than the ERA-15 data. In the 1983-84 season both the ERA-interim and the ERA5 data shows a 12-day run of days with u-
wind greater than 2 m/s with 5 days of OLR below 220 w/m2. If only two days within this spell did not meet the 2 m/s threshold in the ERA-15 data, then the next monsoonal burst, which occurred on 5 January 1984, would have been counted as the onset date, as was shown in Hung and Yanai (2004). In the 1989-90 season the westerly winds reached the 2 m/s threshold on 6 January (the onset date for that season from Hung and Yanai, 2004) but dropped below 2 m/s on the 7th and then above again on the 8th through the 14th, making only a seven-day run, and then above the threshold again from the 14th through the 31st. On
the days below the threshold, the winds are still westerly and are close to 2 m/s. It is quite possible that the ERA-15 data maintained a strong enough burst to show a 10+ day run beginning on the 6th.

The extended Hung and Yanai (2004) dataset, with the outliers retained, shows a mean onset date of 27 December with a standard deviation of 20 days.

**Davidson et al. (2007)** report the mean onset date is 2 January. The reproduced dataset captured the precise onset dates as the
original dataset in only four of the 15 seasons analysed by Davidson et al. (2007), it was off by only one day for 6 of the 15 seasons, and off by more than one day but less then 7 days three of the 15 seasons and more than 1 week different for 2 seasons (1989-90 and 1990-91, see Figure 1g). The extended dataset shows a mean monsoon onset of 2 January with a standard deviation of 17 days.

**Zhang (2010)** original onset dates and the recreated onset dates are compared in Figure 1h. Of the 43 years considered by
Zhang (2010), we were able to successfully recreate the precise onset date of only 15 (35%) of the years, and within 3 days for 36 seasons (84%). Of the remaining 6 seasons, the recreation differed from the original dates by one to almost 6 weeks, with the largest difference of -39 days in the 1985-86 season. Because of the large differences in these seasons, we do not consider this to be a successful reproduction. It is not clear what caused the differences, although we found the analysis to be very sensitive to the period selected for the climatological mean $PW_{max}$ and $PW_{min}$ (i.e. whether the $PW_{max}$ and $PW_{min}$ were
calculated over the full wet season or just the monsoon months). Due to these large discrepancies for more than 10% of the re-created dataset and the limitations with the ERA-40 reanalysis data (mentioned in Section 3.9) we chose not to calculate an extended dataset (see Figure 1h) and Zhang (2010) data are not included in the correlations calculations (Section 4.3).

For the **Kajikawa et al. (2010)** reproduction, we were able to successfully re-create the daily AUSMI values, but found some discrepancies when applying the onset criteria. By adjusting the threshold of the third criteria listed in Section 3.8 we were





able to find a closer match for most years. Of the 58 years included in the original study, we were able to reproduce 52 seasons (90%) to within three days of the original dates including 20 onset dates matched the Kajikawa et al. (2010) dates precisely (see Figure 1i). Two of the onset dates were more than three days but less than 1 week different and four had a more than seven days but less than two weeks difference between the original and reproduced onset dates. Kajikawa et al. (2010) noted a mean onset data of 15 December with a standard deviation of 16 days. The re-created dataset shows the same statistics for the same years, but when using the extended dataset, the standard deviation is 15 days.

## 4.2 Comparison with climate indices

Here we show the full analysis of statistics for the Drosdowsky (1996) onset methodology. We will then report the results of applying the same methodology to the other monsoon onset datasets.

Drosdowsky (1996) reported a correlation coefficient between onset date and the September-November (SON) SOI of $\rho=-0.56$ for the period of 1957 through 1992. However, when this dataset is extended to 2021 the correlation drops to $\rho=-0.40$ ($p<0.05$), and when analysing only the extended data (i.e.1992 to 2021) the correlation is even lower ($\rho=-0.23$). Using an arbitrary threshold of a seasonal SON SOI value of +/- 7 to define the ENSO state (i.e. values > +7 indicate a La Niña state and values < -7 indicate an El Niño state) we can see that in the original dataset there were eight El Niño years and six La Niña years. In the latter part of the dataset there were seven more El Niño years and seven more La Niña years with some strong La Niña events (SOI >10, or one standard deviation of MSLP anomalies) that are not present in the earlier part of the record.

When considering sea surface temperatures, rather than the SOI, the correlations are equally small. The correlation coefficient between the monsoon onset dates and ENSO indices are in Table 3. The onset dates showed the highest correlation with the SON NINO4 index ($\rho=0.37$), however, that correlation weakens when the background warning trend is removed from the SST index. Both the NINO3.4 and the ENSO modoki index showed correlation coefficients of 0.31. When filtering out neutral years (NINO3.4 anomaly >-0.7 and <0.7) the correlation coefficient increases to $\rho=0.40$, and when filtering out all neutral and weak events (NINO3.4 index within 1 std) the correlation coefficient increases to $\rho=0.43$ ($p=0.06$).

The delay in onset during strong El Niño years is small compared to the expedition of onset dates during strong La Niña years, +2 days and -14 days, respectively (see Figure 2c). Figure 2a shows the Drosdowsky (1996) onset dates for each season from 1950/51 to 2020/21 with each season shaded by weak/strong and +/- ENSO state based on the NINO3.4 index. In Figure 2b the data has been sorted by onset date and in Figure 2c the data has been grouped by ENSO state. The same analyses for the SOI, NINO3, NINO4 and ENSO_modoki show similar patterns as the NINO3.4 analysis shown in Figure 2. Our results show that only a strong La Niña has a meaningful impact on Australian monsoon onset timing at Darwin using the Drosdowsky (1996) onset method. Using the 70 years in this study as a basic statistical model, there is a 60% probability that onset will be delayed given an El Niño with SON NINO3.4 anomaly > 0.98° C (1 standard deviation) and an 81% probability that onset will be early given a La Niña with SON NINO3.4 anomaly < -0.98° C, compared with a 48% probability when using all data.



(The three seasons that showed a delayed onset dates in La Niña years were calculated using pre-1957 Darwin sounding data and had irregular timing and reported at irregular heights).

The correlations with non-ENSO climate drivers listed in Table 2 showed mixed results. Most of the correlations did not show statistical significance. Only the IOD, the Western North Pacific Monsoon Index (WNPMI) index and the Indonesian Index

SST (II SST; Verdon and Franks, 2005) showed statistical significance, yet the correlations were relatively low.

The detrended IOD showed positive correlations of over 30% for the month of September ($\rho$=0.37) and for the season of August-October (ASO; $\rho$=0.34). When the trends were retained the correlations were similar. A positive IOD has a stronger tendency toward delaying the monsoon onset than a negative IOD has in expediting the onset. Of the 70 years considered, 83% of the monsoon onset's during a positive IOD were delayed while only 58% of the onset dates during a negative IOD were

early. There were two years when the monsoon onset occurred in November (1973, 2013), these were both neutral IOD years, suggesting the IOD was not a factor in the early onset. Nearly half of the onset dates occurred in December, although the IOD pattern has usually diminished by the time the monsoon has begun; there is no statistically significant link between the December IOD index and December onset dates.

The June–September WNPMI showed a correlation coefficient with monsoon onset timing of $\rho$=0.37 (p<0.05). The II SST

showed a small negative correlation with Drosdowsky (1996) onset dates. The mean SST for the June–August had a correlation of $\rho$=$-$0.24 (p<0.05) which gradually increased to $\rho$=$-$0.26 (p<0.05) in the SON season. The present study found that the correlation between the June–September Indian monsoon index and the Drosdowsky (1996) monsoon onset dates were low and lacked statistical significance.

### 4.3 Correlation analysis

Following the same process as was followed to test the correlations with the Drosdowsky (1996) onset dates, here we show the results of the correlations with the other extended datasets.

For **Troup (1961)**, neither the wind nor the rainfall onset criteria timing showed a significant correlation with any ENSO, IOD or SST indicators. The extended dataset showed statistically significant, albeit small, correlations with high latitude variability, specifically the Amundsen Sea Low Index and a delayed correlation with the Antarctic Oscillation. The Troup (1961) rainfall

onset correlated with the AAO for the seasons August-September ($\rho$=0.33) and September-November ($\rho$=0.31) as well as the October-December mean Relative Central Pressure variation of the Amundsen Sea Low Index ($\rho$= -0.37). The Troup (1961) wind onset correlated with the longitude variation of the Amundsen Sea Low Index for the seasonal mean of June-August ($\rho$= -0.40), July-September ($\rho$= -0.30), and August-October ($\rho$= -0.41), and the latitude variation of the Amundsen Sea Low Index for the seasonal mean of September-November ($\rho$=0.37), October-December ($\rho$=0.45). The physical mechanism causing this



relationship should be the topic of future research. The combined Troup (1961) wind and rainfall criteria did not show a statistically significant relationship with any climate indices.

**Murakami and Sumi (1982)** showed a statistically significant correlation with the detrended NINO4 index for the season JAS (ρ=0.32). However, when the trend is retained the relationship weakens such that only the September-November season shows a relationship with a correlation greater than 30% (ρ=0.33). Murakami and Sumi (1982) also showed a statistically

significant correlation with the ENSO modoki index, the NINO3 and NINO3.4 SST indices, but in all cases the correlation coefficient is less than ±30%. When considering strong ENSO events, where the seasonal index exceed ± one standard deviation, the correlations with the Murakami and Sumi (1982) onset dates increased. For the detrended SON NINO3.4 index the correlation coefficient increased from ρ=0.23 to ρ=0.40. Of the 70 years considered, 70% of the monsoon onset's during a strong El Niño were delayed while 90% of the onset dates during a strong La Niña were early. We also considered the

correlations with non-ENSO climate drivers listed in Table 2. Most of the correlations did not show statistical significance. Only the detrended IOD indices showed statistically significance, but the correlations were small, i.e. within ±30%. When retaining the trend in the IOD pattern, the correlation coefficients were lower in every case, suggesting that what little correlation exists between the Murakami and Sumi (1982) monsoon onset dates and the IOD is diminishing over time. When the IOD is not present or the DMI is neutral, it is not a factor in driving monsoon onset timing, the correlation (τ) is near 7%

and onset dates range from 1 December to 10 February. However, when considering only the events when the detrended SON IOD is not neutral (seasonal average DMI is <-0.4 or >+0.4), the correlation coefficient with onset dates increases to τ=0.43with p<0.05. This relationship is stronger for positive IOD events than for negative. Nine of 10 events when the SON mean DMI was greater than +0.4, the onset was delayed. Only 1997 showed a positive IOD event with an early onset, and this onset date came only 4 days before the long-term average. Of the eight negative IOD events, six showed an early onset. This pattern

breaks down when the trend is retained, where the probability of a delayed onset for neutral, negative and positive events are 49%, 58% and 75%, respectively. Thus, we conclude that when using Murakami and Sumi (1982) onset criteria, a positive IOD is likely to delay the monsoon onset while a neutral and negative IOD have little to no impact on onset timing.

**The Holland (1986)** reported "no significant correlation" between SOI values prior to onset and monsoon onset timing. The extended dataset also shows a lack of significant correlation with the SOI, or any other ENSO index, or the IOD or any index

used in this study. Due to the lack of statistical significance and the low correlations coefficients, we conclude that the Holland (1986) method of monitoring the onset of the Australian monsoon has low predictability on a seasonal time scale.

The re-created **Hung and Yanai (2004)** extended data set showed a statically significant correlation with the ENSO modoki indices, the WNPMI and the seasonal IOD indices, but all the correlations were small, i.e. within ±30%. The Hung and Yanai (2004) monsoon onset dates showed the highest correlation with the detrended monthly September IOD index with ρ=0.30

(p<0.05)





The extended **Davidson et al. (2007)** onset dates show only a very weak correlation with ENSO indices. Of all the correlations with p<0.05, only three showed any correlation > ±30% and are the detrended seasonal JAS NINO4 (ρ=0.34); detrended seasonal JJA NINO3.4 (ρ=0.31), and the JAS seasonal average Amundsen Sea Low relative central pressure index showed a correlation of ρ=0.35.

The extended **Kajikawa et al. (2010)** monsoon onset dataset showed the strongest link with seasonal-scale climate indices of all the monsoon onset datasets examined here. These correlations are listed in Table 4 Correlation coefficients of seasonal climate indices with Kajikawa et al. (2010) monsoon onset dates. Only |ρ|>0.3 and p<0.05 are shown. Kajikawa et al., (2010) reported correlation coefficients of -0.48 for onset dates and the SOI during November and December. Our recreated onset dates correlate with the SOI for November with ρ=-0.50 and for December ρ=−0.48. When the datasets are extended to the 450 2020 season the correlation coefficients become ρ=−0.49 and ρ=−0.44 for November and December, respectively, and ρ=−0.48 for the seasonal SON mean SOI.

**5 Discussion and conclusions**

Drosdowsky (1996) calculated a correlation of ρ=−0.56 with September-November (SON) mean SOI. We found the same correlation when considering the same time period (1957-1992), but the correlation lowered to ρ=−0.40 when the dataset is 455 extended from 1950-2021. We suppose two possible explanations for this change: (1) the initial sample size was too small to correctly capture the climate variability (and may still be), and/or (2) background trends in climate patterns are changing the link between the onset and the SOI in the months before the onset (beyond testing for trends in the data, these theories are untested and may be a topic of future research).

We then tested the correlation with the extended Drosdowsky (1996) onset dates and other ENSO indices. Correlation 460 coefficients with NINO3, NINO4, NINO3.4 and ENSO_modoki indices all showed statistical significance, but the correlation values were all low with the highest correlation being the SON NINO4 index with ρ=0.37. When using the statistical correlation as an indicator of possible predictability (i.e. > 60% correlation) we found that none of the ENSO indices showed a strong link with Australian monsoon onset timing at Darwin using the Drosdowsky (1996) onset method. When not considering the statistical correlation, but simply analysing onset dates by ENSO state, we found that only a strong La Niña had a meaningful 465 impact on monsoon onset timing, suggesting a non-linearity in the relationship between ENSO and monsoon onset. The monsoon onset was earlier than the average for eight out of ten strong La Niña years, compared to five out of 10 ENSO-neutral years and four out of ten El Niño years.

We also considered the correlations with non-ENSO climate indices in seasons before the monsoon onset. Climate influences from the previous season that do not correlate well with the timing of the Drosdowsky (1996) onset dates include the 470 stratospheric QBO, polar annular modes, the PDO, the Indian monsoon in the previous season, the Amundsen Sea low, and



Indian Ocean SST. The monthly September and October IOD, and the seasonal average ASO and SON IOD, measured by the Dipole Mode Index (DMI), showed a weak (30-40%) correlation. The IOD pattern usually dissipates before the monsoon onset in late December or early January (Saji et al., 1999), as does the correlation with the DMI and onset dates in the OND season and the individual months, November and December. When isolating IOD states and then comparing with onset dates, it

appears that a positive IOD tends to delay onsets more than negative IOD expedites onset, suggesting a non-linear relationship.

When considering other onset definitions of the dynamical monsoon onset, neither the Troup (1961) combined wind and rain index, Holland (1986), nor Hung and Yanai (2004) extended onset dates showed a statistically significant correlation larger than ±30% with ENSO variability, Indian Ocean SST or any other climate indices considered in this study. Overall, these monsoon onset methods lacked a relationship with largescale climate patterns. The Holland (1986) Australian monsoon onset

definition was especially problematic when also considering the difficulty in recreating the methodology.

The extended Kajikawa et al. (2010) dataset showed correlations with ENSO indices that were similar to the extended Drosdowsky (1996) when all the data was considered, but when considering only strong ENSO events the Drosdowsky (1996) data showed a stronger relationship with the seasonal NINO3.4 indices while the Kajikawa et al. (2010) correlations showed little change. Overall, both the Drosdowsky (1996) and Kajikawa et al. (2010) methods provided insight to the monsoon

dynamics and some level of predictability with seasonal-scale climate patterns.

The extended Murakami and Sumi (1982) onset dataset showed statistically significant, but low (<30%) correlations with the IOD and ENSO indices. These correlations changed when removing neutral ENSO and IOD? events from the analysis, specifically, a positive SON mean DMI is often associated with a delayed monsoon onset while a neutral and negative SON mean DMI have no relationship with onset timing. This onset criterion was relatively easy to calculate and use, and could be

included with the Drosdowsky (1996) and Kajikawa et al. (2010) methodologies as one that provides some prognostic capabilities.

The relationships between the SOI and monsoon onset dates that were reported in Drosdowsky (1996) weaken when the dataset is extended to include the monsoon seasons from 1950-51 through 2020-21. When considering other ENSO indices, only a strong La Niña (e.g., SON NINO3.4 index > 0.98 °C) has an impact on monsoon onset timing, where eight of ten strong La

Niña events were associated with an expedited monsoon onset. The extended Murakami and Sumi (1982) onset dataset and the extended Kajikawa et al.,(2010) dataset showed similar relationships, although the correlations with Murakami and Sumi (1982) showed smaller correlations and the Kajikawa et al.,(2010) did not show differences between strong and neutral ENSO events.

When considering the influence of other climate patterns on the monsoon onset dates, the seasonal and monthly detrended

DMI showed similar correlations as the ENSO indices with the Drosdowsky (1996), Murakami and Sumi (1982) and Kajikawa

et al.,(2010) onset methodologies. However, these were small to moderate correlations (<±40%) which diminish as the IOD pattern breaks down (usually in December). Also, when the trend in the IOD data is retained the relationship with onset dates diminishes in most (but not all) cases.

To conclude, the relationship between ENSO and Australian monsoonal variability has been heavily studied, with most studies pointing to a positive correlation. However, we have shown that the timing of the dynamical monsoon onset is one aspect of variability that does not show a strong link with ENSO or other seasonal-scale indices. We have also shown that the relationship with some of these indices is non-linear, with a strong La Niña showing a stronger influence than a strong El Niño and a strong positive IOD in the season leading up to onset tends to have a stronger influence than a negative onset. We have also shown that the already-weak relationships between onset timing and the IOD and ENSO are weakening over time, but we have not assessed if this weakening is due to simply more data capturing a larger breadth of the climate variability or if the background warming trend in sea surface temperatures is changing the physical relationship between the climate pattern and the monsoon. Other global monsoon patterns, such as the Indian and the Southeast Asian monsoons show a similar link with ENSO; could similar analysis of onset timing further our understanding of these monsoon patterns? Future research should look at the linkages to other monsoon patters and the teleconnections and other physical relationships linking these climate drivers with onset dates. Another question for future research is, while statistical relationships are weak, could dynamical models predict the onset of the monsoon on seasonal time scales?

**Data availability**

All data analyzed in this research are available via the URLs provided in Table 1 and Table 2.

**Author contributions**

JL designed the study, carried out the analysis underpinning the paper, and wrote the draft manuscript. JR advised JL throughout this work, contributed to the interpretation of the results and to the writing of the manuscript.

**Competing interests**

The authors declare that they have no conflict of interest.

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



## Figures



Figure 1 Re-created and extended monsoon onset dates using the methods described in each respective paper.



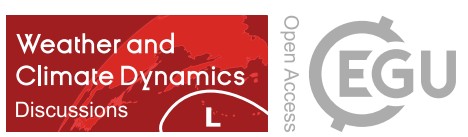

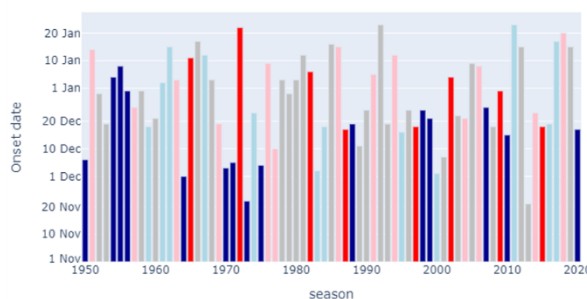

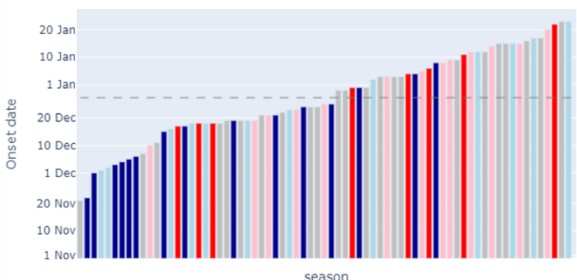

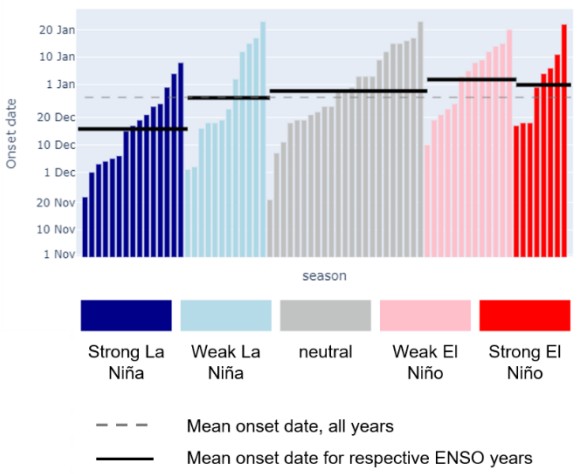

**Figure 2 Analysis of Drosdowsky (1996) onset dates with NINO3.4 SST anomalies. Colours are as follows:** *dark blue* = SST < -0.99 °C; *light blue* = -0.99 °C ≤ SST < -0.5 °C; *grey* = -5 °C ≤ SST ≤ +0.5 °C; *pink* = +0.5 °C < SST ≤ +0.99 °C; *red* = SST > +0.99 °C. **(a) Monsoon onset dates for each season, coloured by NINO3.4 SST anomalies. (b) Monsoon onset dates sorted by onset date and coloured by NINO3.4 SST anomalies. (c) Monsoon onset dates grouped by ENSO state and coloured by NINO3.4 SST anomalies.**




**Tables**

Table 1: Data used to calculate Australian monsoon onset dates. The web links listed in the table were valid as of the date this work was submitted.





| Reference | Original Data used in reference | Data used in re-creation |
|---|---|---|
| Troup (1961) | Rainfall from 6 locations near Darwin and Darwin sounding data | Rainfall data from the SILO dataset https://www.longpaddock.qld.gov.au/silo/point-data/<br><br>Darwin sounding data |
| Murakami and Sumi (1982) | A custom-made data set prepared of twice-daily, 2.5 deg resolution zonal wind data at 850 hPa (see Sumi and Murakami 1981) | NCEP/NCAR 40-year reanalysis data (Kalnay et al., 1996) http://www.cdc.noaa.gov/ |
| Holland (1986) | Daily averaged Darwin Airport sounding | Daily averaged Darwin Airport sounding |
| Drosdowsky (1996) | Darwin Airport sounding | Darwin Airport sounding |
| Hung and Yanai (2004) | Wind data from 15-year (1979-93) ECMWF reanalysis (ERA-15) project data (J. K. Gibson, P. Kållberg, S. Uppala, A. Hernandez, A. Nomura, 1999)<br><br>OLR data source was not described | Wind data from ECMWF ERA5 reanalysis<br><br>https://www.ecmwf.int/en/forecasts/datasets/reanalysis-datasets/era5<br><br>(Copernicus Climate Change Service (C3S), 2017)<br><br><br>OLR data was obtained from www.ncei.noaa.gov (Hai-Tien & NOAA CDR Program, 2011) https://www.ncdc.noaa.gov/<br><br>cdr/<br><br>atmospheric/<br><br>outgoing-longwave-radiation-daily |





| Davidson et al. (2007) | NCEP/NCAR 40-year reanalysis data (Kalnay et al., 1996) http://www.cdc.noaa.gov/ | NCEP/NCAR 40-year reanalysis data (Kalnay et al., 1996) http://www.cdc.noaa.gov/ |
| Kajikawa et al. (2010) | NCEP/NCAR 40-year reanalysis data (Kalnay et al., 1996) http://www.cdc.noaa.gov/ | NCEP/NCAR 40-year reanalysis data (Kalnay et al., 1996) http://www.cdc.noaa.gov/ |

Table 2 Data sources for major climate drivers used in this study. The web links listed in the table were valid as of the date this work was submitted.

| Index | | Data Source web address as of January 2021 |
|---|---|---|
| ENSO | NINO 3 <br> NINO 3.4 <br> NINO 4 | ERSST5 data from <br><br> https://www.cpc.ncep.noaa.gov/data/indices/ |
| | El Niño modoki index | http://www.jamstec.go.jp/frsgc/research/d1/iod/modoki_home.html.en |
| | Monthly Southern Oscillation Index (SOI) | http://www.bom.gov.au/climate/current/soi2.shtml <br><br> (Troup, 1965) |
| Indian Ocean (SSTs) | Indian Ocean Dipole (IOD) Mode Index (DMI) | https://psl.noaa.gov/gcos_wgsp/Timeseries/DMI/ <br><br> (Saji et al., 1999) |
| | Indian Ocean Basin-Wide SST index | ERSST.v5 data averaged from 25 N to 25 S, 30 E to 120 E. <br><br> (adapted from Taschetto et al., 2011) |
| | Indonesian Index Sea Surface Temperature (II SST) | ERSST.v5 <br><br> (Verdon and Franks, 2005) |
| | Indian monsoon index (IMI) | http://apdrc.soest.hawaii.edu/projects/monsoon/seasonal-monidx.html <br><br> (Wang et al., 2001; Wang and Fan, 1999) |





| Northern Hemisphere Monsoons | Webster and Yang Monsoon Index (WYM) | http://apdrc.soest.hawaii.edu/projects/monsoon/seasonal-monidx.html <br> (Webster and Yang, 1992) |
|---|---|---|
| | Western North Pacific Monsoon index (WNPMI) | http://apdrc.soest.hawaii.edu/projects/monsoon/seasonal-monidx.html <br> (Wang et al., 2001; Wang and Fan, 1999) |
| Polar Annular modes | Arctic Oscillation (AO) | https://www.ncdc.noaa.gov/teleconnections/ao/ <br> (Dai and Tan, 2017; Thompson and Wallace, 1998) |
| | Antarctic Oscillation (AAO) | https://www.cpc.ncep.noaa.gov/products/precip/CWlink/daily_ao_index/aao/aao.shtml <br> (Mo, 2000) |
| | Marshall, station based Southern Annular Mode index (SAM) | https://climatedataguide.ucar.edu/climate-data/marshall-southern-annular-mode-sam-index-station-based <br> (Marshall and (Eds)., 2018) |
| Quasi-Biennial Oscillation (QBO) | | Standardised U 50 hPa winds <br> https://www.cpc.ncep.noaa.gov/data/indices/ |
| North Atlantic Oscillation (NAO) | | https://www.ncdc.noaa.gov/teleconnections/nao/ <br> (Barnston and Livezey, 1987) |
| Amundsen Sea Low index (ASLI) | Latitude variations | https://climatedataguide.ucar.edu/climate-data/amundsen-sea-low-indices <br> (Raphael et al., 2016) |
| | Longitude variations | |
| | Relative central pressure | |






Table 3 Correlation Coefficients of seasonal ENSO indices with Drosdowsky 1996 monsoon onset dates. Only statistically significant (p<0.05) values are shown.

| Correlations with Drosdowsky (1996) | | | | | |
|---|---|---|---|---|---|
| | Index | | | | |
| Season | NINO3 | NINO3.4 (detrended) | NINO4 (detrended) | SOI | Modoki index |
| JJA | ρ = 0.25 | ρ = 0.32 (0.33) | ρ = 0.32 | ρ = -0.25 | |
| JAS | τ = 0.17 | ρ = 0.29 (0.30) | ρ = 0.34 (0.38) | ρ = -0.29 | τ = 0.20 |
| ASO | τ = 0.16 | ρ = 0.29 (0.30) | ρ = 0.35 | ρ = -0.33 | τ = 0.27 |
| SON | τ = 0.17 | ρ = 0.31 (0.31) | ρ = 0.37 (0.23) | ρ = -0.40 | τ = 0.31 |


Table 4 Correlation coefficients of seasonal climate indices with Kajikawa et al. (2010) monsoon onset dates. Only |ρ|>0.3 and p<0.05 are shown.

| Correlations (ρ) with Kajikawa et al. (2010) | | | | | |
|---|---|---|---|---|---|
| | Index | | | | |
| Season | NINO3.4 (detrended) | NINO4 (detrended) | SOI | IIsst | Detrended IOD |
| JJA | 0.36 (0.41) | 0.34 (0.47) | -0.37 | -0.35 | |
| JAS | 0.34 (0.39) | 0.33 | -0.41 | -0.38 | 0.30 |
| ASO | 0.34 (0.38) | 0.34 | -0.42 | -0.43 | 0.37 |
| SON | 0.36 (0.40) | 0.36 | -0.48 | -0.43 | |