# Peer review of "Seasonal climate influences on the timing of the Australian monsoon onset"

_Weather and Climate Dynamics, 2021_

## Referee Comment (RC1)

The present study comprehensively explored the relationship between monsoon onset dates and large-scale climate variability by examining a set of Australian monsoon onset indices and climate indices. It is shown that the many of the previously proposed relationships between monsoon onset and climate drivers are less robust considering an extended dataset. The results have significant implications for weather prediction and climate projection. I suppose it is publishable given a few minor edits.

Comments:

1. Figure 1. I found this plot too busy. One way to make it easier to read might be to illustrate the dates using curves and dots (x-axis being time and y-axis being onset date), instead of bars. Also, it'll be interesting (and informative) to group the indices into a couple of groups, for example, indices based on precipitation v.s. circulation, and briefly discuss how these indices compare with each other.
2. Figure 2. Could you please put error bars on the mean?
3. Lines 453-458. The two explanations can be roughly tested, by examining different chunks of data. The current dataset covers 72 years (1950-2021), while Drosdowsky (1996) used a data set of 36 years (1957-1992). (1) The authors could split 1950-2021 into two 36-year subsets and examine the spread between the first subset and latter one. (2) One might be able to obtain some more insights using bootstrapping or other statistical methods.

---

## Author Response (AR1)

We would like to thank both reviewers for their feedback and favourable assessment of our manuscript. All minor comments have been addressed in our response and the manuscript has been modified accordingly. A summary of all the changes is presented in the following table.

| Comment | Response to Reviewer 1 |
|---|---|
| *Figure 1. I found this plot too busy. One way to make it easier to read might be to illustrate the dates using curves and dots (x-axis being time and y-axis being onset date), instead of bars. Also, it'll be interesting (and informative) to group the indices into a couple of groups, for example, indices based on precipitation v.s. circulation, and briefly discuss how these indices compare with each other.* | We would like to thank the reviewer for suggesting an alternative way to summarise and present our results. In generating our Figure 1, we have considered previously several options and now have also followed up on the reviewers suggestion.

Using curves instead of bars as suggested gives the impression that the onset dates are related from one season to the next. However, this requires the reader to know and understand that each onset date is a single point each year that is unrelated to the previous year's onset. We would argue that our 'bar' type presentation avoids this possible misinterpretation by the reader.

In regards to grouping the indices, this grouping was considered in Lisonbee et al. (2020). Thus in doing so again here would reproduce what has been done previously leading to a similar presentation, albeit showing different data. We would argue that our presentation in addition to the description provided in our paper is appropriate in conveying the key information from our analysis.

In summary, we have retained our original presentation of the results (i.e. figure 1) since there appears to be no advantage in following up on the reviewer's suggestion. However, we would like to thank the reviewer for prompting us to check our presentation again. |
| *Figure 2. Could you please put error bars on the mean?* | Yes, thank you for these comments. We modified Figure 2 and have now included a standard deviation bar around the mean. |
| *Lines 453-458. The two explanations can be roughly tested, by examining different chunks of data. The current dataset covers 72 years (1950-2021), while Drosdowsky (1996) used a data set of 36 years (1957-1992). (1) The authors could split 1950-2021 into two 36-year subsets and examine the spread between the first subset and latter one. (2) One might be able to obtain some more insights using bootstrapping or other statistical methods.* | Many thanks for this suggestion. We have followed up on this and added a paragraph, see page 17.

*To roughly test these explanations, we split the data into two periods of 36 years each. A bootstrapping technique was applied to both periods and changes in the data between the two periods were analysed. The mean onset date and STD changed by less than a day between each period. The SOI differs by only 0.4 between the two periods and it is concluded that these changes are small compared to the changes seen in the correlation between the two datasets.* |
| **Comment** | **Response to Reviewer 2** |
| L64-75: Does the dynamical monsoon onset deliver more rain than the start of the wet | Thank you very much for this suggestion. We've added a small statement to our paper. |

| | |
|---|---|
| season? Can't these impacts happen before the dynamical onset? The rainfall-only metrics may be more relevant to monsoon impacts such as floods, water resources and so forth, and may correlate better to climate indices. | Nicholls et al (1982) showed that 30% of the wet season rainfall occurs before the monsoon onset and 70% occurs afterword. At Darwin Airport the average rainfall total on days with > 0.2 mm of rain for non-monsoon days is 14.7 mm while the average daily rainfall amount for monsoon days is 21.3 mm. Pre-monsoonal rainfall is characterized by meso-scale thunderstorms, which may produce large rainfall totals locally on individual days, while monsoonal rainfall can produce large rainfall totals for multiple days and over a very broad areas (Pope et al. 2009). The impacts of wet season rainfall may appear in the early wet season, but the likelihood increases under a persistent monsoon pattern.

It is known that early wet season (Oct-Dec) rainfall correlates well with ENSO (McBride and Nicholls (1983), thus this paper focuses on the dynamical monsoon. |
| L77: It's not clear to me why seasonal scale variability should strongly modulate the timing of onset, which by definition occurs on a sub-seasonal time-scale! MJO, mid-latitude troughs, etc, are all known to be important. Can you expand on why you are trying to do so? Presumably for planning beyond synoptic timescales. | Thank you, this is a really good point. Why should seasonal scale variability influence sub-seasonal scale weather patterns (i.e. the monsoon)? We are investigating this possibility because it has been proposed in previous research (Hendon et al. 1989; Drosdowsky and Wheeler, 2014; Kim et al., 2006; Kullgren and Kim, 2006; Nicholls, 1984; Nicholls et al., 1982; Smith et al., 2008; Webster et al., 1998) and we would like to test these possible connections in the most complete way possible.

The reviewer is correct that the value comes in allowing for planning at longer time scales. At the moment, our best technology for predicting monsoon bursts is the MJO which provides predictability at multi-week timescales. If there was a connection between monsoon onset and seasonal climate drivers it would provide a valuable planning mechanism for agriculture producers and so many others in tropical Australia.

We have re-written the paragraph in question. It now reads:

*The MJO provides predictability of the monsoon onset at multi-week timescales. If there was a connection between monsoon onset and climate drivers on a seasonal (multi-month) time scale it could provide a valuable planning mechanism for agriculture producers and so many others in tropical Australia. This paper presents a statistical analysis of seasonal-scale (1–6 month) climate influences on the timing of the dynamical onset of the Australian monsoon for the period 1950-51 to 2020-21. With the acknowledgement that monsoon onset occurs on a sub-seasonal time scale, we are investigating this possibility because it has been proposed in previous research (Drosdowsky, 1996; Hendon et al., 1989; Kim et al., 2006;* |

| | |
|---|---|
| | *Kullgren and Kim, 2006; Nicholls, 1984; Smith et al., 2008; Webster et al., 1998) and we would like to test these possible connections.* |
| L410: Interesting that the Troup definition lag-correlates reasonably well with the Amundsen Sea Low, and the Antarctic Oscillation. Presumably somehow through mid-latitude variability? See Berry and Reeder study. | Many thanks for this information. We followed up on this and included Berry & Reeder (2016) in our paper |
| The presentation of results could be a little more structured, for example a single table with all onset definitions (rows) and all climate indices (columns), with correlation coefficients in bold text wherever significant might make the paper much more useful as a resource. | Many thanks for this suggestion, which we have addressed this by including a table of correlations with the Sept-Oct-Nov ENSO and IOD indices as exemplary. Although the table shows only a subset of all the correlations tested, it appears apropos considering the focus of the discussion on ENSO and IOD indices in section 5. |